# Acute Metabolic Responses to Glucose and Fructose Supplementation in Healthy Individuals: A Double-Blind Randomized Crossover Placebo-Controlled Trial

**DOI:** 10.3390/nu13114095

**Published:** 2021-11-16

**Authors:** Max L. Eckstein, Antonia Brockfeld, Sandra Haupt, Janis R. Schierbauer, Rebecca T. Zimmer, Nadine Wachsmuth, Beate Zunner, Paul Zimmermann, Barbara Obermayer-Pietsch, Othmar Moser

**Affiliations:** 1Division of Exercise Physiology and Metabolism, Department of Sport Science, University of Bayreuth, 95440 Bayreuth, Germany; max.eckstein@uni-bayreuth.de (M.L.E.); antonia.brockfeld@uni-bayreuth.de (A.B.); sandra.haupt@uni-bayreuth.de (S.H.); janis.schierbauer@uni-bayreuth.de (J.R.S.); rebecca.zimmer@uni-bayreuth.de (R.T.Z.); nadine.wachsmuth@uni-bayreuth.de (N.W.); beate.zunner@uni-bayreuth.de (B.Z.); paul.zimmermann@uni-bayreuth.de (P.Z.); 2Department of Internal Medicine, Division of Endocrinology and Diabetology, Endocrinology Lab Platform, Medical University of Graz, 8036 Graz, Austria; barbara.obermayer@medunigraz.at; 3Interdisciplinary Metabolic Medicine Trials Unit, Department of Internal Medicine, Division of Endocrinology and Diabetology, Medical University of Graz, 8036 Graz, Austria

**Keywords:** glucose, metabolism, fructose, sucralose, healthy individuals, insulin, substrate oxidation

## Abstract

The aim of this study was to investigate the impact of glucose (Glu), fructose (Fru), glucose and fructose (GluFru) and sucralose on blood glucose response in healthy individuals. Fifteen healthy individuals (five females, age of 25.4 ± 2.5 years, BMI of 23.7 ± 1.7 kg/m^2^ with a body mass (BM) of 76.3 ± 12.3 kg) participated in this double-blind randomized crossover placebo-controlled trial. Participants received a mixture of 300 mL of water with 1 g/kg BM of Glu, 1 g/kg BM of Fru, 0.5 g/kg BM of GluFru (each), and 0.2 g sucralose as a placebo. Peak BG values Glu were reached after 40 ± 13 min (peak BG: 141 ± 20 mg/dL), for Fru after 36 ± 22 min (peak BG: 98 ± 7 mg/dL), for GluFru after 29 ± 8 min (BG 128 ± 18 mg/dL), and sucralose after 34 ± 27 min (peak BG: 83 ± 5 mg/dL). Significant differences regarding the time until peak BG were found only between Glu and GluFru supplementation (*p* = 0.02). Peak blood glucose levels were significantly lower following the ingestion of Fru compared to the supplementation of Glu and GluFru (*p* < 0.0001) while Glu and GluFru supplementation showed no difference in peak values (*p* = 0.23). All conditions led to a significantly higher peak BG value compared to sucralose (*p* < 0.0001). Blood lactate increased in Glu (*p* = 0.002), Fru and GluFru (both *p* < 0.0001), whereas sucralose did not increase compared to the baseline (*p* = 0.051). Insulin levels were significantly higher in all conditions at peak compared to sucralose (*p* < 0.0001). The findings of this study prove the feasibility of combined carbohydrate supplementations for many applications in diabetic or healthy exercise cohorts.

## 1. Introduction

Dysglycemia is a frequently discussed topic in science as it is the precursor of metabolic diseases such as obesity, metabolic syndrome, type 1- (T1DM) and type 2 diabetes mellitus (T2DM) [1,2].

It is defined as the absence of euglycemia with a defined blood glucose (BG) range of 70–180 mg/dL and subdivided in hypoglycemia (<70 mg/dL) and hyperglycemia (>180 mg/dL) with harmful changes to physiological states, deteriorating human health by damaging micro- and macrovasculature [3]. In individuals with T2DM, the state of hypoglycemia is not as frequent and common in comparison to individuals with T1DM due to residual hormonal counter-regulation in the case of low levels of BG and exogenous insulin therapy [4]. In individuals with T1DM, exogenous insulin injections are unavoidable, which bears the risk of a glycemic mismanagement, potentially leading to severe hypoglycemic episodes [4]. However, physical activity and extended periods without any supply of carbohydrates may induce hypoglycemia, which requires an immediate consumption of carbohydrates [5,6].

Healthy individuals may also reach lower BG levels, which are considered as physiological immediately counteracted by a hormonal response [7]. However, carbohydrate rich drinks are recommended around strenuous physical activity to fuel performance [8]. In individuals with T1DM, this is also recommended during exercise conditions, not solely to increase performance, but also to reduce the risk of hypoglycemia [9]. Besides general recommendations to consume a certain amount of carbohydrates, it is unclear, whether glucose (Glu), fructose (Fru) or a combination of both (GluFru) increase BG levels faster to maintain euglycemia. The underlying physiological and hormonal pathways during physical exercise in T1DM have been described previously in comprehensive narrative reviews [10,11]. However, the physiological response to supplementation in healthy individuals under resting conditions is still unclear [12].

Therefore, the aim of this study was to investigate the time until reaching peak BG levels following the ingestion of Glu, Fru, and GluFru containing beverages in healthy individuals. Conclusions about the metabolic reaction of the human body to rapid increases in BG was compared to sucralose as a placebo to avoid bias within the cross-over design of the study. Moreover, we investigated the change in substrate oxidation and hormonal response to rapid BG increases for a better understanding of metabolic physiology. 

## 2. Materials and Methods

This was a single center, randomized, double-blind, placebo-controlled crossover clinical trial, assessing the impact of Glu, Fru, GluFru, and sucralose on healthy individuals. The local ethics committee of the University of Bayreuth (Germany) approved the study protocol (O 1305/1.GB, 26 April 2021), which was registered at the German Clinical Trials Register (DRKS00024755). The study was conducted in conformity with the declaration of Helsinki and Good Clinical Practice. Before any trial related activities, potential participants were informed about the study protocol and participants gave their written informed consent. This study is a proof of concept study.

### 2.1. Eligibility Criteria

Eligibility criteria included male or female individuals aged between 18–65 years (with a body mass index (BMI) of 18.0–29.9 kg/m^2^, both inclusive). Participants with a normal glucose tolerance, measured via overnight fasting BG levels, were included. Individuals were excluded if they were enrolled in a different study, received investigational medicinal products, had a supine blood pressure outside of the range of 90–150 mmHg for systolic and 50–95 mmHg for diastolic after resting for five minutes in a supine position. Furthermore, participants were excluded if they had a history of multiple and/or severe allergies to any trial related products.

### 2.2. Assessment of Eligibility

Inclusion and exclusion criteria were assessed by an investigator at the screening visit two weeks prior to the start of the study.

### 2.3. Study Design

After inclusion in the study, participants were assigned to ascending numbers. Participants were then allocated to the order in which the trial visits were conducted in cross-over randomized fashion with the software Research Randomizer^®^ (1:1:1:1) [13]. Participants received 1 g/kg body mass (BM) Glu, 1 g/kg BM Fru, and 0.5 g/kg BM of GluFru (each). Sucralose was given with a fixed amount of 0.2 g per dosage since higher amounts might exert some toxicity. The artificial sweetener was used as a placebo control to imitate the taste of the other study related products to avoid selection bias. Between each visit, a minimum period of 48 h was maintained. 

### 2.4. Trial Visits

Prior the start of each trial visit, participants had to fast for at least 12 h and refrain from any strenuous physical exercise within 24 h prior to each visit. Furthermore, no caffeine rich drinks or diet sodas were allowed within the 12 h fasting periods. Participants were not allowed to consume any alcoholic beverages within 24 h prior to the fasting periods. All participants had to fill in an international physical activity questionnaire in short form (IPAQ-SF) prior to the start of any visit to monitor changes in exercise behavior and to potentially reschedule visits once physical exercise increased to maintain comparability between trial arms.

Participants attended the research facility in the morning after their overnight fast. During the 2-h trial visits, participants remained in a supine position. To measure substrate oxidation, a face mask held in place by a nylon harness covered the participants’ nose and mouth (Metalyzer, Cortex, GER). The mask was attached to a bidirectional digital turbine flow meter to measure the volume of inspired and expired air. A sample line between the turbine and analyzer unit determined O_2_ and CO_2_ content of the air. A two-point calibration procedure was conducted prior to any testing session according to the manufacturer’s guidelines (Calibration Manual 931-00-264/Revision a/2014-03-06, CORTEX Biophysik GmbH, Leipzig, Germany). After a 5-min resting phase, participants were asked to open their spirometry mask and consume a drink of 300 mL water containing either 1 g/kg BM Glu, Fru, GluFru, or 0.2 g sucralose. Participants were asked to consume the drink within one minute and then to close the mask again and to relax. Immediately prior to consuming the drink, a venous blood sample of 8 mL was taken from the antecubital vein to measure fasting insulin, c-peptide, cortisol, glucose, and lactate. For the following two hours of the measurement period, venous BG and lactate samples were measured in a 5-min interval. Hormonal samples were collected at minutes 0, 15, 30, 60, and 120. Spirometric variables were measured via breath-by-breath and averaged for 10 s.

### 2.5. Blood Sampling

Venous BG and lactate samples were collected with 20 µL capillaries from the antecubital vein and analyzed via a fully enzymatic-amperometric method (Biosen S-line, EKF Diagnostics, Barleben, Germany). At timepoints 0, 15, 30, 60, and 120, 8 mL serum blood samples were collected. The blood serum vacutainer was left to rest for a minimum of 30 min prior to being centrifuged at room temperature for 10 min at 1500× *g*. The plasma was then aliquoted and stored at −80 °C at the research facility. Once the study was completed, plasma samples were analyzed by routine clinical biochemistry assays for cortisol, insulin, and c-peptide (Advia Centaur XPT, Siemens, Munich, Germany).

### 2.6. Randomization Procedure

In the morning prior to each visit, a researcher not otherwise involved in the implementation of the trial prepared drinks with Glu, Fru, GluFru, or sucralose, dependent on the randomization, in opaque shaker bottles and labelled them with the participant’s ID. This procedure was conducted to avoid any kind of bias from the researchers or participants by seeing clearness/cloudiness of the drinks prior to ingestion.

### 2.7. Statistical Analysis

All data were assessed for normal distribution by means of the Kolmogorov–Smirnov normality test. Venous BG, lactate, hormones, and spirometric variables were analyzed via the mixed-effects model with Geisser–Greenhouse correction. Differences between groups, timepoints, and group x timepoint were calculated in this fashion. Tukey’s multiple comparisons test with individual variances were computed for each comparison. Peak BG, time until reaching peak BG, peak lactate, and peak hormonal values were calculated via repeated measures one-way ANOVA. Post-hoc tests were performed via Dunn’s test. Statistical significance was accepted at *p* ≤ 0.05.

## 3. Results

In total, 15 healthy individuals (five females) were included in the study with a mean ± SD age of 25.4 ± 2.5 years, BMI of 23.7 ± 1.7 kg/m^2^ with a BM of 76.3 ± 12.3 kg. All screened participants were eligible to participate in the study in which no participant had to be withdrawn or left the study prematurely. 

### 3.1. Blood Glucose

#### 3.1.1. Time to Peak Blood Glucose

Overall, the participants’ peak BG values Glu were reached after 40 ± 13 min (peak BG: 141 ± 20 mg/dL), for Fru after 36 ± 22 min (peak BG: 98 ± 7 mg/dL), for GluFru after 29 ± 8 min (BG 128 ± 18 mg/dL), and sucralose after 34 ± 27 min (peak BG: 83 ± 5 mg/dL). Significant differences were found only between Glu and GluFru supplementation (*p* = 0.02). 

#### 3.1.2. Peak Blood Glucose

BG values were similar at timepoint 0 at 78 ± 5 mg/dL (Glu), 79 ± 6 mg/dL (Fru), 80 ± 8 mg/dL (GluFru), and 80 ± 8 mg/dL (sucralose) (*p* = 0.50). Peak values were significantly higher compared to timepoint 0 with 141 ± 20 mg/dL (Glu), 98 ± 7 mg/dL (Fru) and 128 ± 18 (GluFru) (all *p* < 0.0001). Values for sucralose did not reach significance after changing from baseline until the peak value by only 3 mg/dL (*p* = 0.50). Peak BG levels were significantly lower following the ingestion of Fru compared to the supplementation of Glu and GluFru (*p* < 0.0001) while Glu and GluFru supplementation showed no difference in peak values (*p* = 0.23). All conditions led to a significantly higher blood peak BG value compared to sucralose (*p* < 0.0001) (Figure 1).

No participant reached hyperglycemia defined as ≥180 mg/dL during the course of the study. In each study arm, hypoglycemia (<70 mg/dL) was reached by several participants, within the Glu arm, six participants had hypoglycemia, seven participants in the Fru arm, ten participants in the GluFru arm, and four participants in the sucralose arm. Only in the GluFru study arm did three participants reach a clinically relevant hypoglycemia with values <54 mg/dL. However, no visit had to be cancelled prematurely since participants did not mention any discomfort or hypoglycemic symptoms. 

### 3.2. Blood Lactate

#### 3.2.1. Time to Peak Blood Lactate

No difference in reaching peak blood lactate was found between groups (Glu 70 ± 29 min; Fru 69 ± 20 min; GluFru 61 ± 16 min (*p* = 0.41)).

#### 3.2.2. Peak Blood Lactate

Blood lactate at baseline was 1.00 ± 0.33 mmol/L (Glu), 0.82 ± 0.14 mmol/L (Fru), 0.89 ± 0.25 mmol/L (GluFru), and sucralose 0.89 ± 0.24 mmol/L (sucralose). Significant increases from timepoint 0 until peak blood lactate were noticed for Glu at 1.52 ± 0.25 mmol/L (*p* = 0.002), Fru at 3.06 ± 0.98 mmol/L (*p* < 0.0001), and GluFru at 2.81 ± 0.48 mmol/L (*p* < 0.0001), but not for sucralose at 0.95 ± 0.25 mmol/L (*p* = 0.051). Peak blood lactate values were significantly lower in the Glu arm compared to the GluFru arm (*p* < 0.0001) and Fru arm (*p* = 0.0007). Peak blood lactate with sucralose supplementation was significantly lower compared to all other conditions (*p* < 0.0001). No significant difference in peak blood lactate between Fru and GluFru (*p* = 0.97) was found.

### 3.3. Insulin Response

#### 3.3.1. Time until Peak Insulin Response

Overall, the participants peak insulin values following the ingestion of Glu were reached after 40 [30–50] minutes, for Fru after 25 [20–45] minutes, and for GluFru after 30 [25–35] minutes. No significant difference was found between the groups (*p* = 0.07).

#### 3.3.2. Peak Insulin Response

Baseline insulin values were not significantly different between the different groups (Glu 5.18 ± 2.4 mU/L; Fru 5.15 ± 2.2 mU/L; GluFru 6.2 ± 3.5 mU/L; sucralose 5.5 ± 2.3 mU/L) (*p* = 0.35). Peak insulin values were significantly higher for Glu 38.21 ± 11.70 mU/L (*p* < 0.0001), Fru 13.34 ± 7.11 mU/L (*p* = 0.0055), and GluFru 33.81 ± 13.20 mU/L (*p* < 0.0001) compared to timepoint 0. Insulin levels did not change following the ingestion of sucralose 5.87 ± 2.11 mU/L (*p* = 0.55). Peak insulin levels were significantly higher in Glu compared to Fru (*p* = 0.0002) with no significant difference to GluFru (*p* = 0.39). GluFru levels compared to Fru also demonstrated a significant difference (*p* = 0.003). Peak insulin values were significantly higher in all groups when compared to sucralose (all *p* < 0.0001).

### 3.4. C-Peptide Response

#### 3.4.1. Time until Peak C-Peptide Response

Time until reaching peak c-peptide response for Glu was 52 ± 14 min, for Fru 62 ± 38 min, and for GluFru 44 ± 16 min. There were no significant differences between all study arms (*p* = 0.16).

#### 3.4.2. Peak C-Peptide Response

Baseline c-peptide values were not significantly different between groups (Glu 0.72 ± 0.33 ng/mL; Fru 0.73 ± 0.28 ng/mL; GluFru 0.81 ± 0.30 ng/mL; sucralose 0.70 ± 0.28 ng/mL) (*p* > 0.05). Peak c-peptide levels were significantly higher for Glu 3.39 ± 1.84 ng/mL, Fru 1.65 ± 0.61 ng/mL, and GluFru 2.71 ± 1.06 ng/mL (all *p* < 0.0001) compared to timepoint 0. No significant difference was found for sucralose (*p* = 0.66). Peak c-peptide levels for Glu were significantly higher compared to Fru (*p* = 0.02) but not compared to GluFru (*p* = 0.45). C-peptide levels in GluFru were also higher compared to Fru (*p* = 0.02). All peak c-peptide levels were higher compared to sucralose (*p* < 0.0001).

### 3.5. Cortisol Response

Baseline cortisol values were not significantly different between groups (Glu 152.2 ± 63.6 ng/mL; Fru 144.9 ± 65.5 ng/mL; GluFru 152.9 ± 63.8 ng/mL; sucralose 152.5 ± 52.3 ng/mL) (*p* > 0.05). No change in cortisol levels from timepoint 0 until any timepoint of each trial arm, within and in-between groups was found (*p* = 0.36).

### 3.6. IPAQ-SF

No statistically significant change in physical activity behavior between visits was recorded for each participant (*p* > 0.05). All participants were regularly physically active but avoided any type of demanding exercise within 24 h prior to their study visit. The median activity was 3846 [2703; 4979] Met-mins/week.

**Figure 1 nutrients-13-04095-f001:**
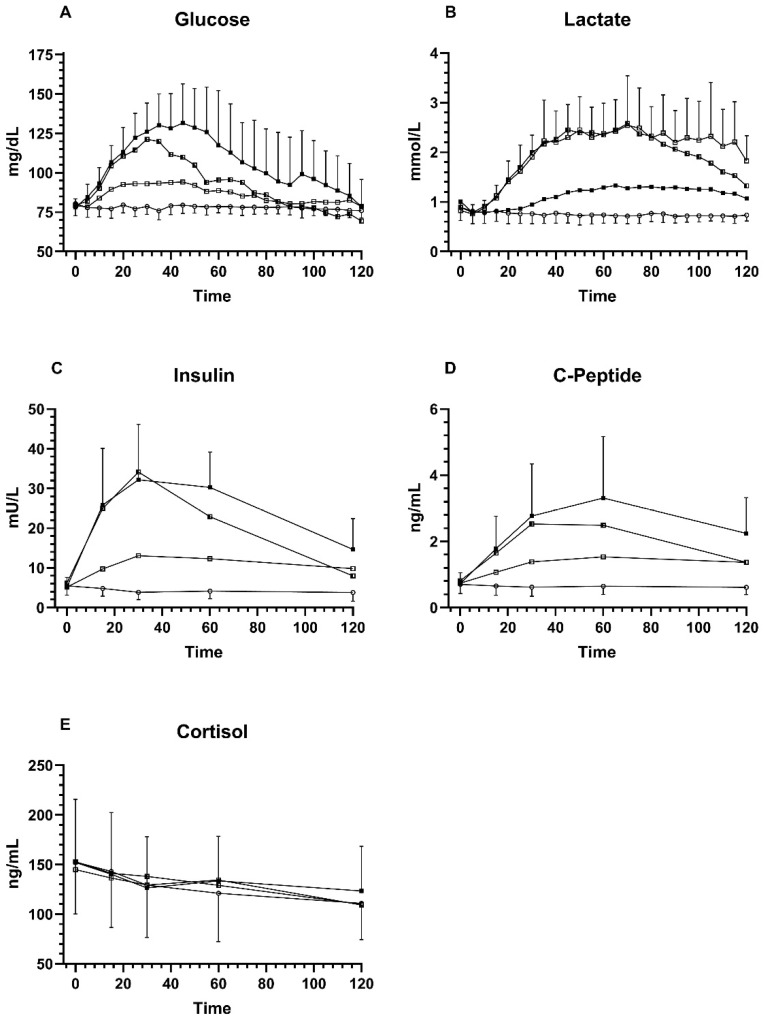
Comparison of Glu, Fru, GluFru, and sucralose supplementation. (**A**) venous blood glucose, (**B**) venous blood lactate, (**C**), serum insulin levels, (**D**) serum c-peptide levels, and (**E**) serum cortisol levels. Black circles indicate glucose, open circles indicate sucralose, open squares indicate fructose, and half squares indicate GluFru.

### 3.7. Respiratory Response

Results of the respiratory data are shown in Table 1. Measurement errors due to talking or inappropriate movement of the mask during the 2-h measurement period were excluded.

### 3.8. Adverse Events

Only one participant felt uncomfortable with low blood pressure (<80/40 mmHg) following the glucose trial arm and remained at the research facility for an additional hour after the end of the visit. Three participants mentioned gastric discomfort following the ingestion of fructose, which was suspected due to the high dose of pure fructose given at once, which did not interfere with the visit procedures.

## 4. Discussion

This proof-of-concept study is the first trial that investigated the impact of glucose, fructose, a combination thereof, and sucralose as a placebo in healthy individuals under resting conditions after an overnight fast. We conclude that GluFru supplementation acts almost similarly on BG levels compared to Glu supplementation alone (Figure 1). This is in contrast to earlier findings from individuals with metabolic diseases with a lower BG and insulin response following the ingestion of glucose and fructose [14].

GluFru supplementation led to an earlier peak BG value when compared to Glu supplementation alone, whereas Glu supplementation lead to a prolonged increase with no significant difference in peak BG values between these two study arms. As previously shown and similar to our results, Fru alone led to a decreased BG increase and a reduced and more modest insulin response compared to glucose. Since Fru does not necessarily need insulin to enter the cell, it bypasses the early rate limiting steps of glucose metabolism, suggesting that a combination of glucose and fructose may lead to a more rapid increase in BG compared to glucose alone, which cannot be directly demonstrated by our results (Figure 1) [15,16]. Even though peak BG was reached faster in GluFru, however, no clear advantage over Glu supplementation can be seen (Figure 1).

After reaching peak BG_,_ GluFru led to a more rapid decrease in BG compared to glucose that maintained higher BG levels and a higher insulin response over time. Fructose has been the subject of debate for several decades in the management of diabetes, mainly T2DM. Increased fructose intake is well known to increase triglyceride levels, facilitate the development of non-alcoholic fatty liver disease, and to deteriorate lipid profiles, in general favoring weight gain and the development of atherosclerosis [17,18]. 

It is well known that lactate is produced during the metabolization of fructose (Figure 1B). From a historical perspective, lactate has turned from a ‘deleterious waste product’ produced during physical exercise or ischemia, to a potential beneficial agent to ‘reshape the field of energy metabolism’ within recent years, according to Rabinowitz et al. [19]. In this context, previous studies have shown that lactate infusions lead to reduced epinephrine responses reducing symptoms of hypoglycemia, shifting glycemic thresholds for these responses to lower BG concentrations, causing brain lactate uptake [20,21]. Lactate is oxidized in the brain and can account for up to 25% of the calculated brain glucose energy deficit, reducing neuroglycopenic symptoms [22]. This coincides with the results from our study, since 10 out of 15 individuals in the GluFru group reached hypoglycemia and three out of 10 reached clinically relevant (yet symptom-free) hypoglycemia, likely due to elevated lactate concentrations. This can also be supported by our findings for cortisol, which showed no significant difference between groups, independent of BG or lactate levels.

Supplementation of Glu, Fru, and GluFru led to a significant increase in carbohydrate oxidation compared to the baseline and returned back to the initial values within two hours. This is different to the findings of Smeraglio et al., who compared glucose and fructose supplementation during mixed meal challenges [23]. However, the amount of fat and protein decelerates and prolongs the uptake of glucose into the blood stream, which in our study was unhindered due to 100% pure glucose, fructose, or a combination thereof. The findings for sucralose led to no changes in substrate oxidation, which was previously shown by Stellingwerf et al. during exercise conditions and underlines the non-calorific artificial sweetening traits of sucralose [24]. Findings from our study are of value for healthy individuals being physically active. Considering the different courses of BG between the substances, GluFru may be a recommendable supplement to enhance and prolong exercise performance by delivering rapid glucose provision into the blood stream, which during exercise is transferred to the cell directly via glucose transporter type 4 (GLUT-4) (glucose) or at rest via GLUT-2 and GLUT-5 (fructose) without the need of insulin secretion and action [12,25]. However, follow-up studies during physical exercise should be conducted to confirm this assumption.

Our study serves as a proof-of-concept study to pave the way for follow-up studies investigating the multifaceted properties of carbohydrate supplementation in health and disease. However, our study has some limitations. In previous studies, investigating long-term supplementation of fructose led to changes in lipid profiles and increased triglycerides. This would also have been of interest to investigate in the course of the study, which was, however, not the aim of this study. Even though our sample size is suitable for a proof-of-concept study, a power-analysis with a higher sample size and a matched sample size for men and women would be helpful to facilitate subgroup analysis, which was not possible due to the lack of data in our study. However, our study serves as a baseline for future projects in delivering evidence suitable for a variety of cohorts for research purposes or in daily life. GluFru may be a favorable type of supplementation compared to Glu alone to stabilize BG levels when in fear of running low in BG. In terms of its potential beneficial effects on exercise performance, future studies should investigate the effects of these carbohydrates and supplementation schemes.

Fructose itself solely leads to a small increase in BG since the majority of it is directly metabolized to lactate, and hence should not be recommended for athletes.

## 5. Conclusions

Glu, Fru, and GluFru increase BG when given as 1 g/kg BM in a liquid form in comparison to sucralose, which had no effect. Glu and GluFru similarly increased BG at almost the same rate, while Fru and GluFru increased lactate similarly without an increased stress response. Our findings prove the feasibility of combined highly-dosed carbohydrate supplementations for many applications in healthy individuals.

## Figures and Tables

**Table 1 nutrients-13-04095-t001:** Respiratory parameters during the 2-h visits.

		Glucose	Fructose	Glucose + Fructose	Sucralose	*p*-Value
VO_2_ (L/min)	Baseline	0.35 ± 0.15	0.45 ± 0.28	0.29 ± 0.15	0.33 ± 0.16	0.17
	Peak	0.49 ± 0.11 *	0.57 ± 0.26	0.40 ± 0.13	0.49 ± 0.13	0.07
	End	0.34 ± 0.07	0.31 ± 0.07	0.32 ± 0.07	0.34 ± 0.07	0.91
VCO_2_ (L/min)	Baseline	0.31 ± 0.12	0.36 ± 0.19	0.27 ± 0.10	0.28 ± 0.12	0.28
	Peak	0.45 ± 0.10	0.53 ± 0.22	0.40 ± 0.07 *	0.43 ± 0.09	0.06
	End	0.32 ± 0.06	0.29 ± 0.06	0.27 ± 0.07	0.29 ± 0.07	0.57
CHO (g/min)	Baseline	0.28 ± 0.10	0.27 ± 0.12	0.17 ± 0.10	0.17 ± 0.11	0.02 ^†^
	Peak	0.57 ± 0.20 *	0.63 ± 0.19 *	0.53 ± 0.09 *	0.30 ± 0.07	0.001 ^†^
	End	0.27 ± 0.15	0.26 ± 0.17	0.17 ± 0.12	0.14 ± 0.10	0.14
Fat (g/min)	Baseline	0.06 ± 0.07	0.10 ± 0.05	0.08 ± 0.05	0.09 ± 0.05	0.46
	Peak	0.14 ± 0.06 *	0.13 ± 0.04	0.12 ± 0.05	0.15 ± 0.05	0.52
	End	0.03 ± 0.03	0.02 ± 0.02	0.05 ± 0.03	0.05 ± 0.03	0.16
RER	Baseline	0.81 ± 0.04	0.86 ± 0.09	0.86 ± 0.09	0.83 ± 0.13	0.38
	Peak	1.02 ± 0.06 *	1.03 ± 0.06 *	1.03 ± 0.05 *	0.93 ± 0.05	0.0002 ^†^
	End	0.93 ± 0.05	0.92 ± 0.06	0.85 ± 0.06	0.85 ± 0.06	0.04 ^†^
BF (1/min)	Baseline	17.6 ± 3.1	16.8 ± 3.8	16.2 ± 5	17.5 ± 6.1	0.77
	Peak	24.1 ± 2.5 *	24.3 ± 3.9 *	22.7 ± 2.2 *	22.4 ± 3.7	0.26
	End	18.1 ± 2.6	17.0 ± 4.2	16.7 ± 2.8	17.6 ± 3.7	0.53
VT (L/min)	Baseline	0.59 ± 0.20	0.60 ± 0.20	0.56 ± 0.22	0.54 ± 0.11	0.54
	Peak	0.85 ± 0.15 *	0.81 ± 0.16 *	0.86 ± 0.11 *	0.71 ± 0.18	0.21
	End	0.65 ± 0.15	0.57 ± 0.16	0.58 ± 0.17	0.55 ± 0.17	0.14
VE (L/min)	Baseline	11.6 ± 3.2	12.8 ± 3.6	9.7 ± 4.2	11.1 ± 3.7	0.20
	Peak	15.1 ± 3.1	15.8 ± 4.3	14.5 ± 2.5 *	14.9 ± 3.1	0.48
	End	10.6 ± 1.5	10.3 ± 2.0	10.1 ± 2.9	10.1 ± 2.8	0.64

VO_2_: Ventilation oxygen. VCO_2_: Ventilation carbon dioxide. CHO: Carbohydrates. RER: Respiratory exchange ratio. BF: Breathing frequency. VT: Tidal volume. VE: Ventilation. * Indicates statistical significance compared to baseline. † Indicates significant treatment effect between groups (*p* < 0.05).

## Data Availability

Data will be made available upon reasonable request by the corresponding author.

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
