# Peer review of "Acute Metabolic Responses to Glucose and Fructose Supplementation in Healthy Individuals: A Double-Blind Randomized Crossover Placebo-Controlled Trial"

_nutrients, 2021, doi:10.3390/nu13114095_

Round 1

Reviewer 1 Report

General comment

In its simplicity the experiment is interesting and useful. The method is correct, the same for the analysis. The weakness is in the discussion: results are debated in healthy, in diabetes and in athletes, but there aren’t enough elements for deductions so enlarged. It would be better to concentrate only in the kind of enrolled population.

Reviewer 2 Report

The reviewer acknowledge the authors for this manuscript and supports that a good description of the scope of the study as well as a good bibliography search was performed. the authors have evaluated important para meters for the problematic of sugar supplementing in healthy people going or not for exercise. minor corrections must be made:

Introduction line 37: type 2 diabetes is missing in the sentence "type 1- (T1DM) and mellitus (T2DM)"

Materials and methods line 116 the abbreviations have been already described previously in the section

results line 166: "blood glucose" was already described and abbreviated as BG previously in the text
